# Heart Rate Variability and Stress Recovery Responses during a Training Camp in Elite Young Canoe Sprint Athletes

**DOI:** 10.3390/sports7050126

**Published:** 2019-05-23

**Authors:** André B. Coelho, Fábio Y. Nakamura, Micaela C. Morgado, Clifton J. Holmes, Angela Di Baldassarre, Michael R. Esco, Luis M. Rama

**Affiliations:** 1Research Center for Sport and Physical Activity (CIDAF), Faculty of Sports Science and Physical Education, University of Coimbra, Coimbra 3040-256, Portugal; luisrama@fcdef.uc.pt; 2Department of Human Resources Training, Portuguese Canoe Federation, Vila Nova de Gaia 4430-750, Portugal; 3Department of Medicine and Aging Sciences, “G. d’Annunzio”, University of Chieti-Pescara, Chieti - Via dei Vestini 31, Italy; fabioy_nakamura@yahoo.com.br (F.Y.N.); a.dibaldassarre@unich.it (A.D.B.); 4The College of Healthcare Sciences, James Cook University, Townsville QLD 4811, Australia; 5Faculty of Sports, University of Porto, Porto 4200-450, Portugal; micaelacmorgado@hotmail.com; 6Department of Kinesiology, The University of Alabama, Tuscaloosa, AL 35487, USA; cjholmes2@crimson.ua.edu (C.J.H.); mresco@ua.edu (M.R.E.)

**Keywords:** canoeists, cardiac autonomic function, psychometrics, training impulse

## Abstract

Training camps are typical in elite Canoeing preparation, during which, the care to assure adaptation to avoid undesired fatigue is not always present. This study aimed identifying a specific sex response in perceived training loads, recovery and stress balance, and cardiac autonomic responses. Twenty-one elite athletes (11 males and 10 females) of the Portuguese Canoeing National team participated in the investigation. The daily HRV (lnRMSSD) was monitored. The (RESTQ-52) questionnaire was used to access the recovery and stress state. The 10-day training camp was composed of two consecutive 5-day periods (P1 and P2). Data analyses were performed using confidence limits, effect size, and magnitude-based inference. In the females, Session rating of perceived exertion (sRPE), lnRMSSD, and its coefficient of variation did not change between P1 and P2. However, in males, lnRMSSD showed a small reduction from P1 to P2. Also, sRPE was higher in males over the training period, with a possibly small difference at P2. Regarding RESTQ-52, total stress most likely increased with large and very large differences in males and moderate differences in females during the training period. Male canoeists undertook higher perceived training loads than females, with a consequent higher level of total perceived stress and lnRMSSD during a 10-day training camp.

## 1. Introduction

In sports such as canoe sprint it is typical to organize training camps aiming to optimize performance and select the best athletes to compete in international events. In some cases, training load can be intensified during training camps [1]. Thus, coaches need to control stress and recovery level of athletes in order to avoid nonfunctional overreaching [2], which can increase the risk of injury and infection without providing a “supercompensation” effect to performance [3]. Therefore, development of simple, noninvasive monitoring tools that identify early fatigue accumulation is an important area within sport science. However, there is limited evidence relating to the usefulness of practical monitoring tools to control and refine the training of canoeing athletes, especially in elite youth samples.

Session rating of perceived exertion (sRPE) is a valid method to quantify internal training load in canoe sprinters due to its high correlation with heart rate-based training impulse [4]. In other sports, sRPE has been shown to be sensitive to changes in external training load when training is intended to induce overreaching and performance decrement [2,5,6]. On the other hand, within optimal ranges, sRPE accumulated during soccer preseasons is positively correlated with changes in intermittent performance tests (e.g., Yo-Yo intermittent recovery) [7]. However, in young canoe sprinters, the typical training loads undertaken during crucial phases of the preparation (i.e., National Team training camps) leading up to team formation and its effects on stress markers are not documented. These typical values and the associated intersubject variability need to be better understood to provide references to coaches and sports scientists managing the training of young canoe sprint athletes.

The analysis of stress and recovery states during different preparation phases can be performed using a multilevel approach involving psychometric responses [8]. The Recovery-Stress Questionnaire for Athletes (RESTQ-Sport) is a practical and valid psychometric tool for monitoring self-reported responses to training [9]. The psychometric assessment of competitive rowers during intensified training periods suggests the existence of a dose–response relationship between training volume and somatic components of stress and recovery [9]. In soccer, the prolonged accumulation of stress and reduced recovery lead to a decrement in performance in adult players [10]. The reported perceived imbalance is also related to reduced cardiac vagal activity in judo athletes during exposure to high training loads [11]. The RESTQ-Sport has not been used in young canoe athletes, so its sensitivity to the sport-specific loading schemes has not been investigated. Also, female basketball players present lower recovery levels in some items of RESTQ-Sport than males [12]. However, differences between sexes in individual sports are under-reported.

Resting heart rate variability (HRV) has become one of the most widely used objective monitoring tools among athletes [13]. It is accepted that subtle changes in HRV (i.e., substantial increase or decrease coupled with negative changes in perceptual variables) is a sign of excessive overload, which can impair performance and training adaptation [10,14]. For these purposes, the natural logarithm of the mean square difference of the successive R-Wave time instants intervals (RR)(lnRMSSD) is considered the preferred HRV metric for tracking maladaptation and performance impairment [13]. For this, at least three weekly data points should be recorded for reliable indication of a microcycle’s “average” vagal activity [15]. On the other hand, the coefficient of variation of lnRMSSD (lnRMSSD_CV_) can be also calculated weekly to reflect the day-to-day variations in cardiac parasympathetic activity, providing valuable information concerning the training-induced perturbation of homeostasis [16]. Athletes with lower lnRMSSD_CV_ are more aerobically fit and seem to cope better with a given level of training load by reporting reduced perceived fatigue [17]. However, it remains to be investigated whether lnRMSSD and lnRMSSD_CV_ are useful markers to monitor young canoe sprinters of both sexes during training camps. 

Because the lack of information regarding elite youth canoeists, the two main aims of this study were: first, we sought to identify possible sex differences in perceived training load, recovery and stress balance, and cardiac autonomic responses in elite young canoe sprint athletes during a 10-day training camp. Second, we wanted to determine whether loading pattern during the training camp can be identified in this sample of athletes based on the perceptual and HRV responses.

## 2. Materials and Methods

### 2.1. Participants 

Twenty-one elite young athletes (11 males age: 16.21 ± 0.62 years; height: 173.3 ± 5.4 cm; body mass: 68.5 ± 5.0 kg, and 10 females; age: 16.01 ± 0.76 years; height: 166.0 ± 6.1 cm; body mass: 61.1 ± 7.4 kg) from the Portuguese National Canoeing Team (canoe sprint) took part in this study. All the athletes had at least 4 years of experience in competitive canoeing. After receiving the detailed explanation of the purposes and procedures of the study, the legal guardians and the athletes provided their written informed consent. The local Ethics Committee of the Faculty of Sports Science and Physical Education, University of Coimbra (CE/FCDEF-UC/00292019) approved the procedures. 

During the competitive phase a 10-day training camp was scheduled. This planned period was critical because the National Team was selected to finely prepare for the Olympic Hopes, based on the achievement of predefined times in different race distances. The observation period ended 23 days before the start of the competition. Hence, the study period comprised regular training without deliberate intensification to induce functional overreaching and tapering was undertaken after the study, beginning approximately 15 days before the Olympic Hopes. In this competition, the Portuguese National Canoeing Team participated in 13 finals and won three silver medals.

### 2.2. HRV Calculation 

The RR-interval recordings were obtained in a comfortable back straight-seated position, with open eyes and spontaneous breathing. A portable heart rate monitor (Polar Team2, Polar, Kempele Finland) was used to record RR-intervals for 2 min continuously. The first minute was discarded (stabilization period), while the second 1 min period was used to calculate lnRMSSD [18], through Kubios HRV Standard 3.0.0 software (Kubios, Kuopio, Finland). This ultrashort-term HRV determination is considered valid [19] and reliable [20] in sports settings. The time and the place of data collection were always the same, and athletes were asked to consume a similar breakfast and avoid stimulants (coffee, tea, and alcohol) during the study duration. This procedure was repeated every morning during the 10-day observation period, between breakfast and 30 min before the first training session of the day. The mean lnRMSSD value for each 5-day period was recorded (lnRMSSDM). Additionally, we calculated the lnRMSSDcv by the following equation; $lnRMSSDcv=\left(\frac{SD}{lnRMSSD{}_{M}}\right)*100$, where lnRMSSD_M_ corresponds to the mean value of the period during which CV was calculated (5 days). All female athletes informed that their last menstrual period started 7–9 days before the beginning of the study, which corresponds to the ovulatory phase (end of follicular and the beginning of luteal phases).

### 2.3. Training Load

For quantification of session training load (TL), we used the sRPE. This procedure consisted of asking to the athletes to rate their level of perceived exertion using the CR100 [21] scale 15–30 min postexercise. This perceptual value was multiplied by the session duration in minutes [22]. The sRPE results were divided by 10 to allow for comparison with previous studies that used the CR10 scale [23,24,25]. The 10-day training camp was separated into two 5-day periods (P1 and P2) to better analyse the dynamic of training and adaption throughout the overall period. 

### 2.4. Recovery-Stress Questionnaire-52 (RESTQ-52)

The RESTQ-52 [23] was administered on the 1st (M1), 5th (M2), and 10th (M3) day of the training camp. It was fully explained before the assessment and every doubt clarified was clarified by one the researchers. The RESTQ-52 is a modified version of the RESTQ-76, with easier application due to the lower number of questions and has been reported as an appropriate tool to assess the perceived state of recovery and stress in athletes. The questionnaire had 52 questions with Likert-scale responses anchored from 0 (never) to 6 (always) for the athlete to indicate how often she/he participated in various activities during the past 3 days and nights. The questionnaire has 12 scales that assess various stressing agents (general stress and general recovery activities) and recovery agents and 7 additional sports-specific scales, with 4 questions per scale. The scores of the stress-related scales were added and divided by the number of scales to obtain a total stress score. The same procedure was used for the recovery-oriented scales, resulting in a total recovery score. This questionnaire has been widely used in previous studies [24]. 

### 2.5. Statistical Analyses 

The data are presented as mean ± standard deviation (SD) and 90% confidence limits (CL). A specific spreadsheet (xPostOnlyCrossover.xls) was used to compare (within-sex) the sRPE, lnRMSSD_M_, lnRMSSD_CV_, and RESTQ-52 in the different moments. The sRPE, lnRMSSD_M_, lnRMSSD_CV_, and RESTQ-52 were also analysed between sexes using another spreadsheet (xCompare2groups.xls). The third spreadsheet (xParallelGroupsTrial.xls) was used to determine the standardized differences of changes in means between-group in all variables and across the moments. The standardized differences (i.e., effect sizes or ES [25]) were rated using thresholds described by Hopkins et al. [26]: 0–0.2 was trivial, 0.2–0.6 was small, 0.6–1.2 was moderate, 1.2–2.0 was large, >2.0 was very large. The effect was deemed unclear when the CL crossed the threshold for both substantially positive (0.2) and negative (−0.2) values [27]. The quantitative changes for the aforementioned comparisons were qualitatively rated as follows; <1%, almost certainly not; 1% to 5%, very unlikely; 5% to 25%, unlikely; 25% to 75%, possible; 75% to 95%, likely; 95% to 99%, very likely; >99%, almost certain [26]. If the change of higher or lower differences was >5%, then the true difference was assessed as unclear [27]. The RMSSD were first log-transformed (natural logarithm) to reduce bias arising from nonuniformity error.

## 3. Results

The ten days workout routine and training loads of both males and females are presented in Table 1. The ten days had the same pattern in all training sessions, warm-up, main part and cool-down. During this period, competition distances (1000 meters (m), 500 m, and 200 m) and strength training (resistance and power) were prioritized, but training of the general endurance component (running) was also done.

Table 2 displays the sRPE and heart rate variability variables (lnRMSSD_M_ and lnRMSSD_CV_) in male and female canoeists across the training periods (P1 and P2). When comparing sexes, there was a trend towards higher sRPE values in males when compare to females over the training period, with observation of a *possibly small* difference in P2. The within-group analyses revealed that the female athletes did not show differences in the sRPE, lnRMSSD_M_ and lnRMSSD_CV_ between P1 and P2. In the male athletes group the lnRMSSD_M_ showed a likely small reduction from P1 to P2. 

Figure 1 displays the RESTQ-52 scores of the total stress and total recovery for male and female athletes in the three timepoints (M1, M2, and M3). Regarding the responses to RESTQ-52 Sport, in males, total stress *most likely* increased with *large* and *very large* differences between M1 and M2, and between M1 and M3, respectively. Total recovery increased across the three moments (*possibly and likely small* effects). In females, total stress *most likely* increased from M1 to both M2 and M3, with *moderate* effect. Similar to males, females increased total recovery across the three moments (*possibly* and *likely small* effects). Between sexes, total stress was *likely* higher (*small* effect) in males than in females in M3.

## 4. Discussion

This study was primarily designed to identify possible sex differences in perceived training loads, recovery, and stress balance and cardiac autonomic responses in elite young canoe sprint athletes during a training camp. Secondly, the study sought to determine whether loading pattern during training camps can be identified based on perceptual and HRV responses. There were several noteworthy findings. First, young elite-level canoeists appear to accumulate between 2740 to 3024 au within five days of a training camp that is designed to select athletes for the National Team. Second, compared to the female, the male athletes perceived higher training loads and display greater levels of total perceived stress at the end of the 10-day training camp. Third, the male athletes demonstrated a trend towards a decrement in lnRMSSD_M_ at the second period of the training camp (P2) when compared to the initial period (P1), while the female athletes showed no change in HRV between the two periods.

Responses in HRV appear to be linked to changes in external training load and training-related stress in several studies. For instance, Flatt and Esco [14] suggested that lnRMSSD_CV_ may be a suitable marker for reflecting the acute adjustment of weekly training load in female team sport athletes. In addition, Le Meur et al. [28] found small weekly mean changes in the lnRMSSD (ES = 0.40) after overload training that resulted in functional overreaching in elite endurance athletes. Moreover, overload training was associated with a reduction in lnRMSSD_M_ along with an increase in lnRMSSD_CV_, concurrent with decrements in perceived fatigue and muscle soreness in Division-1 collegiate sprint-swimmers preceding competition, and was reversed with tapering [29]. The lack of differences in lnRMSSD_CV_ between P1 and P2 found in the current study could be related to the lower daily fluctuation of the weekly training load that was scarce and did not seem to perturb homeostasis.

The male athletes displayed no meaningful change in training loads between P1 and P2; however, they showed higher perceived training loads values than the females in P2 (*possibly small*). As both sex groups performed the same external load, the higher personal commitment to training of the males could explain this result. Though lnRMSSD_CV_ has been suggested as a marker for gauging the challenges of training load to homeostasis [29], the lack of differences between training periods (P1 and P2) confirm the small level of exertion imposed along the training camp.

The male athletes presented a *likely negative* change in lnRMSSD_M_ between P1 and P2, which is possibly a reflex of a higher perceived training load by males, over the entire duration of the training camp, when compared to the females. Of note, the lnRMSSD_M_ difference between P1 and P2 was higher (5.2%) than the smallest worthwhile change (3%) described in the literature to produce a physiological adaptation in males [16]. This difference was also observed in the RESTQ-52 at M3, where the athletes presented *likely* higher total stress than females. Hence, it appears that lnRMSSD_M_ response and perceived stress responses are coherently linked in this study, as previously shown in a study with female soccer players [14].

In the current study we did not find any changes in either lnRMSSD_M_ or lnRMSSD_CV_ between P1 and P2 in the female athletes. However, this finding is similar to previous research involving female soccer players [30]. Weekly lnRMSSD_M_ and lnRMSSD_CV_ presented predictable dose–response relationship with training load as assessed by sRPE. The weeks with higher perceived training loads resulted in lower lnRMSSD_M_ and higher lnRMSSD_CV_, respectively, when compared to those with lower perceived training loads [30,31]. Hence, upon unchanged training loads between consecutive short-term training periods in the current study, the finding of unchanged cardiac vagal activity was expected. Interestingly, both total stress and total recovery assessed by RESTQ-52 increased from M1 to M2 and M3 in both sex groups. This finding suggests that the exertion of the overall training camp was not excessive since, during these periods of overload, the stress increase was concomitant to the decrease in recovery [9,11].

There were some limitations related to the control of extraneous cofactors that could influence the studied variables. For instance, HRV measurements were taken every morning following the breakfast and 30 min before the first training session. Although the athletes were encouraged to maintain a similar breakfast pattern and to avoid stimulants during the 10 days, the amount and type of food consumed were not recorded for each athlete and could have affected HRV measurements. In the female athletes, the menstrual cycle was controlled. This observational research study was specifically undertaken to monitor athletes during their training practices across the 10-day training camp period, excluding the potential interferences of daily practices when they stay at club base. Observational investigations such as the current study are needed to better understand the signs of overreaching and recovery process. Moreover, the 10-day observational period was too short for gauging chronic responses. Baseline data of weeks of limited training are essential to analyse the changes in another period. Therefore, longitudinal research needed to determine if HRV is an adequate procedure for objectively tracking chronic changes, recovery status and adaptation in elite youth canoeist and similar athletes, constituting important and novel grounds for further research in this sport and providing support for credible methodologies. In the canoeing training, sometimes it was complicated for the athletes to wait for more than 15 min at the end of the training in order to collect the RPE data since there was the risk of being wet and getting sick. Data collection was then done after the shower, which corresponded to a 15–30 min interval, with most of the data collection being done at 30 min post-training.

Finally, it is important to recognize that the magnitude-based inference statistical approach used in the current study is being debated in the literature [32]. Nevertheless, reporting estimates of the effect and their uncertainty is considered an adequate research practice to allow researchers and practitioners to decide upon the meaningfulness and practical relevance of scientific findings.

## 5. Conclusions

In conclusion, the male canoeists undertook higher external and internal training loads during National Team training camps. This difference was small but translated into meaningful changes in the vagally-mediated HRV that decreases during the period only in males. Accordingly, total stress assessed by RESTQ-52 was higher at the end of the training camp in males than in females. The monitoring of HRV and stress recovery balance can help to control the undesired effects of excessive loading and other added stressors in the period leading up to important competitions, in both male and female athletes.

## Figures and Tables

**Figure 1 sports-07-00126-f001:**
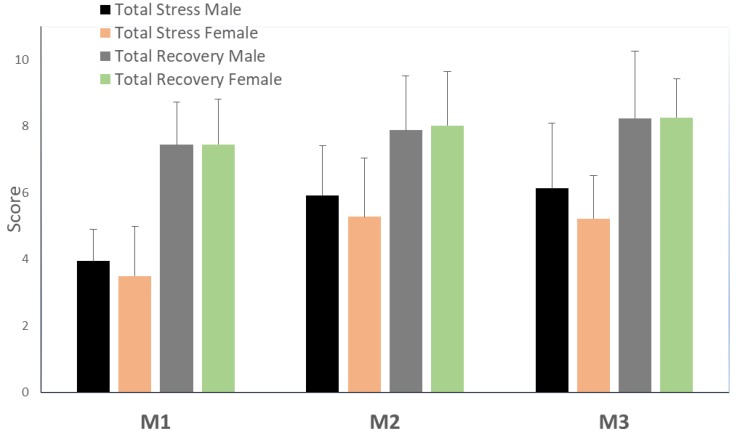
Mean and SD of the RESTQ-52, values of total stress, and total recovery in male and female athletes. M1 (First day of training camp); M2 (Fifth day – middle of training camp); M3 (tenth day – Last day of training camp)

**Table 1 sports-07-00126-t001:** General description of training content (tasks) and daily training load (in arbitrary units; mean ± SD) during each of the Period 1 and Period 2 (P1 and P2) male and female athletes.

**P1**			
**Day 1**			
**Male morning**	**Female morning**	**Male afternoon**	**Female afternoon**
on-water Tasks: 10 km—SR 70–75 sRPE: 2450 ± 1227	on-water Tasks: 10 km—SR 70–75 sRPE: 1893 ± 868	strength Tasks: 6 Exercises, 5 sets, 8 reps with 80%, rest 2’ sRPE: 5844 ± 1570	strength Tasks: 6 Exercises, 4 sets, 8 reps with 80%, rest 2’ sRPE: 4959 ± 1204
**Day 2**			
**Male morning**	**Female morning**	**Male afternoon**	**Female afternoon**
on-water Tasks: (5 × 1’/2’) SR 95–100 + (5 × 45’’/2’) SR 105 sRPE: 6298 ± 668	on-water Tasks: (5 × 45’’/2’) SR 95–100 + (5 × 30’’/2’) SR 105 sRPE: 4744 ± 1432	run Tasks: 40 minutes sRPE: 3015 ± 386	run Tasks: 40 minutes sRPE: 2680 ± 569
**Day 3**			
**Male morning**	**Female morning**	**Male afternoon**	**Female afternoon**
on-water Tasks: 5 × 2’/3’ SR 85 sRPE: 3996 ± 1489	on-water Tasks: 4 × 2’/3’ SR 85 sRPE: 3129 ± 1360	strength Tasks: 5 Exercises, 3 sets, 8 reps with 75%, rest 2’ sRPE: 1811 ± 774	strength Tasks: 5 Exercises, 3 sets, 8 reps with 75%, rest 2’ sRPE: 1560 ± 746
**Day 4**			
**Male morning**	**Female morning**	**Male afternoon**	**Female afternoon**
on-water Tasks: 8 km SR 55–60 sRPE: 1175 ± 676	on-water Tasks: 8 km SR 55–60 sRPE: 1012 ± 719	rest	rest
**Day 5**			
**Male morning**	**Female morning**	**Male afternoon**	**Female afternoon**
on-water Tasks: 1 × 500 m 100% + 8 km SR 55–60 sRPE: 5089 ± 1918	on-water Tasks: 1 × 500 m 100% + 8 km SR 55–60 sRPE: 6417 ± 705	strength Tasks: 4 × (10 exercises, 20 rep, with 40%, rest 20’’)/3’ sRPE: 3019 ± 656	strength Tasks: 3 × (10 exercises, 20 rep, with 40%, rest 20’’)/3’ sRPE: 3033 ± 613
**P2**			
**Day 6**			
**Male morning**	**Female morning**	**Male afternoon**	**Female afternoon**
on-water Tasks: 2 × 1000 m SR 90 + 2 × 750 m SR 105 + 3 × 100 m SR 110 + 2 × 200 m SR 130 sRPE: 5425 ± 853	on-water Tasks: 2 × 500 m SR 95 + 2 × 350 m SR 110+ 3 × 100 m SR 110 + 2 × 200 m SR 130 sRPE: 4832 ± 1124	on-water Tasks: 6 km SR 60 sRPE: 1916 ± 628	on-water Tasks: 6 km SR 60 sRPE: 1900 ± 699
**Day 7**			
**Male morning**	**Female morning**	**Male afternoon**	**Female afternoon**
on-water Tasks: 8 km SR 65 + 5 × 10’’/1’ sRPE: 1707 ± 709	on-water Tasks: 8 km SR 65 + 5 × 10’’/1’ sRPE: 1644 ± 589	rest	rest
**Day 8**			
**Male morning**	**Female morning**	**Male afternoon**	**Female afternoon**
on-water Tasks: 3 × 100 m/6’ SR maximum + 3 × 200 m /6’ SR 120 sRPE: 4930 ± 999	on-water Tasks: 3 × 100 m/6’ SR maximum + 3 × 200 m /6’ SR 120 sRPE: 4110 ± 1133	strength Tasks: 5 Exercises, 5 sets, 8 reps with 80%, rest 2’ + run 30’ sRPE: 8302 ± 1832	strength Tasks: 5 Exercises, 5 sets, 8 reps with 80%, rest 2’ + run 30’ sRPE: 7728 ± 2394
**Day 9**			
**Male morning**	**Female morning**	**Male afternoon**	**Female afternoon**
on-water Tasks: 3 × 1000 m/ 6’ SR 90 + 3 × 750 m/6’ SR 110 sRPE: 6245 ± 415	on-water Tasks: 3 × 500 m/ 6’ SR 95 + 3 × 350 m/6’ SR 115 sRPE: 4275 ± 1281	on-water Tasks: 6 km SR 60 sRPE: 862 ± 567	on-water Tasks: 6 km SR 60 sRPE: 1280 ± 451
**Day 10**			
**Male morning**	**Female morning**	**Male afternoon**	**Female afternoon**
on-water Tasks: 3×(30’’+45’’+1’+1’30’’+1’+45’’+30’’)/1’30’’ SR 110-100-90-80-90-100-110 sRPE: 5104 ± 1008	on-water Tasks: 3×(15’’+30’’+45’’+1’+45’’+30’’+15’’)/1’30’’ SR 120-110-100-90-100-110-120 sRPE: 3825 ± 921	rest	rest

Repetitions: reps; Stroke Rate (SR) (strokes per minute); Meters (m); ‘(minutes); “(seconds).

**Table 2 sports-07-00126-t002:** Session rating of perceived exertion (sRPE) and heart rate variability variables (lnRMSSD_M_ and lnRMSSD_CV_) in male and female canoeist across two training periods (P1 and P2).

				Differences		
	Variable	P1	P2	Standardized Difference (90% CL)	Chances	Qa
Male	sRPE	3024 ± 543	2982 ± 473 *	−0.07 (−0.23–0.08)	0.5/91.4/8.1%	Likely trivial
	lnRMSSD_M_	3.46 ± 0.52	3.28 ± 0.46	−0.32 (−0.63–0.01)	0.6/25/74.4%	Likely negative
	lnRMSSD_CV_	10.53 ± 4.34	8.71 ± 6.38	−0.39 (−1.20–0.42)	10.9/23.2/65.9%	Unclear
Female	sRPE	2753 ± 493	2740 ± 499	−0.02 (−0.13–0.09)	0.2/98.9/0.9%	Very likey trivial
	lnRMSSD_M_	3.38 ± 0.27	3.42 ± 0.23	0.12 (−0.36–0.60)	38.3/49/12.7%	Unclear
	lnRMSSD_CV_	10.29 ± 3.46	9.75 ± 6.34	−0.14 (−1.24–0.96)	29.1/24.7/46.2%	Unclear

* Meaningful difference between sexes in P2 (Male > Female) (ES: 0.23), Qa possibly negative.

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
