# Peer review of "Heart Rate Variability and Stress Recovery Responses during a Training Camp in Elite Young Canoe Sprint Athletes"

_sports, 2019, doi:10.3390/sports7050126_

Round 1
Reviewer 1 Report
The authors should be commending on a well written and well thought out paper.
I have no major concerns regarding the methods and results of the manuscript.
The authors have done a thorough job of framing the study and used somewhat appropriate tools to identify and potential differences between sexes.
It may be worth incorporating a RM ANOVA approach to further support the results found as MBI has been shown to have limitations. Either this or mention potential limitations of the statistical approach in the conclusion.
Author Response
Comments and Suggestions for Authors
The authors should be commending on a well written and well thought out paper.
We thank the reviewer this comment. We agree, the manuscript was revised and improved, and we hope that now follows a correct written English pattern.
I have no major concerns regarding the methods and results of the manuscript.
We thank the reviewer this comment.
The authors have done a thorough job of framing the study and used somewhat appropriate tools to identify and potential differences between sexes.
We thank the reviewer this comment.
It may be worth incorporating a RM ANOVA approach to further support the results found as MBI has been shown to have limitations. Either this or mention potential limitations of the statistical approach in the conclusion.
We thank the reviewer this comment. However, we used this statistical analysis procedure because it has been widely used in HRV studies. The parametric test does not consider the magnitude differences of the effect (Hopkins (2009)). These statistical inference procedures were used in several manuscripts dealing with HRV in the sports field:
Plews, D. J., Laursen, P. B., Stanley, J., Kilding, A. E., & Buchheit, M. (2013). Training adaptation and heart rate variability in elite endurance athletes: Opening the door to effective monitoring. Sports Medicine, 43(9), 773-781.
Flatt, A. A., Esco, M. R., Nakamura, F. Y., & Plews, D. J. (2017). Interpreting daily heart rate variability changes in collegiate female soccer players. J Sports Med Phys Fitness, 57(6), 907-915. doi:10.23736/s0022-4707.16.06322-2
Esco, M. R., Flatt, A. A., & Nakamura, F. Y. (2016). Initial Weekly HRV Response is Related to the Prospective Change in VO2max in Female Soccer Players. Int J Sports Med. doi:10.1055/s-0035-1569342
Nakamura, F. Y., Flatt, A. A., Pereira, L. A., Ramirez-Campillo, R., Loturco, I., & Esco, M. R. (2015). Ultra-Short-Term Heart Rate Variability is Sensitive to Training Effects in Team Sports Players. JOURNAL OF SPORTS SCIENCE & MEDICINE, 14(3), 602.

Reviewer 2 Report
The manuscript was worked out properly (methodology, preparation of research results, selection of literature).
I suggest change the key words. “Flatwater kayak” is not the most important in this manuscript. Rather canoeists (elithe athletes).
Abstract: there is lack of introductory sentence (first sentence). I don`t understand why the Authors wrote “likely small”, “most likely”? Either there are differences at the 0.05 significance level, or there are none. According to me, there should be an unambiguous record. Alternatively, you can write about "tendencies".
Materials and method: The Authors should write separately age and anthropometric information male and female. They was researched two groups, not one.
Discussion: The Authors used terms: “likely negative”, “likely” – I don’t think that these terms should be used in research article. What does mean “likely negative”?
In this section, the authors should also explain (try) why there were differences between male and female.
Conclusions: They require a supplement to explain why the differences between male and female occur.
Author Response
Comments and Suggestions for Authors
The manuscript was worked out properly (methodology, preparation of research results, selection of literature).
We thank the reviewer this comment.
I suggest change the key words. “Flatwater kayak” is not the most important in this manuscript. Rather canoeists (elithe athletes).
We thank the reviewer this comment, and we change de Key word.
Abstract: there is lack of introductory sentence (first sentence).
We thank the reviewer this comment, and we revised the Abstract, and introduce a paragraph before the objectives.
I don`t understand why the Authors wrote “likely small”, “most likely”? Either there are differences at the 0.05 significance level, or there are none. According to me, there should be an unambiguous record. Alternatively, you can write about "tendencies".
We thank the reviewer this comment. We agree that most interesting results found in the study are highlighted through the optional statistical procedure. The statistical analysis used in this study was conduct through calculation of the confidence limits (90%), the effect size and the magnitude-based inference, according to Hopkins (2009) and Batterham & Hopkins (2006).
Materials and method: The Authors should write separately age and anthropometric information male and female. They were researched two groups, not one.
The reviewer remark is right, and we change the manuscript adding anthropometric data separated by sex.
Discussion: The Authors used terms: “likely negative”, “likely” – I don’t think that these terms should be used in research article. What does mean “likely negative”?
We thank the reviewer. However, we analysed the magnitude of the differences between time points and sex based on Effect Size calculation of the mean differences, the 90% of the Confidence interval and the Magnitude Based Inference procedures as it was present in the statistical section of the Methods. According the confidence interval (90%) was assessed qualitatively as follows: <1%, almost certainly not; 1% to 5%, very unlikely; 5% to 25%, unlikely; 25% to 75%, possible; 75% to 95%, likely; 95% to 99%, very likely; >99%, almost certain (Batterham & Hopkins (2006); Hopkins (2009).
These statistical inference procedures were used in several manuscripts dealing with HRV:
Plews, D. J., Laursen, P. B., Stanley, J., Kilding, A. E., & Buchheit, M. (2013). Training adaptation and heart rate variability in elite endurance athletes: Opening the door to effective monitoring. Sports Medicine, 43(9), 773-781.
Flatt, A. A., Esco, M. R., Nakamura, F. Y., & Plews, D. J. (2017). Interpreting daily heart rate variability changes in collegiate female soccer players. J Sports Med Phys Fitness, 57(6), 907-915. doi:10.23736/s0022-4707.16.06322-2
Esco, M. R., Flatt, A. A., & Nakamura, F. Y. (2016). Initial Weekly HRV Response is Related to the Prospective Change in VO2max in Female Soccer Players. Int J Sports Med. doi:10.1055/s-0035-1569342
Nakamura, F. Y., Flatt, A. A., Pereira, L. A., Ramirez-Campillo, R., Loturco, I., & Esco, M. R. (2015). Ultra-Short-Term Heart Rate Variability is Sensitive to Training Effects in Team Sports Players. JOURNAL OF SPORTS SCIENCE & MEDICINE, 14(3), 602.
In this section, the authors should also explain (try) why there were differences between male and female.
We agree with the reviewer, we have amended the text to clarify this point, hopefully it has become clearer.
Conclusions: They require a supplement to explain why the differences between male and female occur.
We thank the reviewer the opportunity to clarify the discussion of this result. The difference founded between the sexes can be attributed to the general commitment or attitude facing the training challenge, with the males reporting higher perceived RPE, which could explain the differences in the trend of decrease in lnRMSSD concomitant with the intensified perception of the internal training load.

Reviewer 3 Report
The data in this study are potentially interesting, but they are ill-described and interpreted. Generally, the manuscript is obscure and difficult to follow. Since it is just a descriptive study, a better explanation regarding why is important and the practical applications of the analysis should be clearly stated.
Affiliations
Please amend them. They are wrong in the current state.
Abstract
Please respect the word count.
Introduction
The introduction is not contextual regarding canoeing. It should be strengthened.
To analyze the loading pattern with just two points of averaged measures seems less than optimal.
Methods
Carefulness regarding the clearance period when collecting HR is limited.
There are differences in sRPE when reporting 15 or 30 min post-exercise. It also seems a lack of carefulness when collecting sRPE. Additionally, raw data regarding sRPE would be interesting.
Day-by-day modulation of the variables would be interesting to report.
Additionally to MBI, a frequentist analysis is needed.
Results
At the current moment, results do not seem to offer a lot of insight regarding canoe training. They should be better presented.
RESTQ-52 analysis should be presented.
Training workload is not a result.
Discussion:
There is an abyss between results and interpretation. Data should be better explained and described, and then better interpreted. There is a lack of context also in discussion.
The aims of the study do not seem to be answered properly.
Author Response
Comments and Suggestions for Authors
The data in this study are potentially interesting, but they are ill-described and interpreted. Generally, the manuscript is obscure and difficult to follow. Since it is just a descriptive study, a better explanation regarding why is important and the practical applications of the analysis should be clearly stated.
We thank the reviewer, the text has been revised, which we hope will have satisfied the reviewer's doubts.
Affiliations
Please amend them. They are wrong in the current state.
We thank the reviewer, we check the affiliations and correct.
Abstract
Please respect the word count.
We thank the reviewer, we correct the count.
Introduction
The introduction is not contextual regarding canoeing. It should be strengthened.
We thank the reviewer; the main objective of this research is to use strategies that can help the coaches and particular canoeing coaches on the use monitoring adaptation tools to the training load.
To analyze the loading pattern with just two points of averaged measures seems less than optimal.
We agree and was assumed as a limitation (in final of discussion).
Methods
Carefulness regarding the clearance period when collecting HR is limited.
We thank the reviewer, and we agree, but this was an observational research investigation. Indeed, it was specifically undertaken to monitor the group of athletes during their standardized training practices across the 10-day period, without the potential interferences to daily practices that may come with experimental, well-controlled designs (this investigation preceded the most important international competition, being a very sensitive period). Observational investigations such as the current study are needed to better understand how to collect and interpret HRV in field settings. Moreover, the HRV measurements were taken every morning following the consumption of breakfast and 30 minutes before the first training session. The amount and type of food consumed was not recorded for each individual athlete and could may have affected HRV measurements, although the athletes were encouraged to maintain a similar breakfast and to avoid stimulants during the 10 days.
There are differences in sRPE when reporting 15- or 30-min post-exercise. It also seems a lack of carefulness when collecting sRPE. Additionally, raw data regarding sRPE would be interesting.
We thank the reviewer this comment. We understand the suggestion, but in the real world of canoeing training, sometimes it is complicated for the athletes to wait for more than 15 minutes at the end of the training. There is the risk of being wet and get sick, so we chose to collect the RPE data after the shower, which corresponded to the interval 15-30 minutes, thinking about the health and well-being of the athletes. However, we realized that it would be desirable to maintain the 30 minutes, which occurred most of the occasions.
Day-by-day modulation of the variables would be interesting to report.
We thank the reviewer this comment. We agree that it could be interesting, however, we intend to minimize data noise, avoiding excess data and facilitating the authors reading.
Additionally, to MBI, a frequentist analysis is needed.
We thank the reviewer.
We used this statistical analysis because it has been widely used for HRV studies. The statistical inference procedures were used in several manuscripts dealing with HRV:
Plews, D. J., Laursen, P. B., Stanley, J., Kilding, A. E., & Buchheit, M. (2013). Training adaptation and heart rate variability in elite endurance athletes: Opening the door to effective monitoring. Sports Medicine, 43(9), 773-781.
Flatt, A. A., Esco, M. R., Nakamura, F. Y., & Plews, D. J. (2017). Interpreting daily heart rate variability changes in collegiate female soccer players. J Sports Med Phys Fitness, 57(6), 907-915. doi:10.23736/s0022-4707.16.06322-2
Esco, M. R., Flatt, A. A., & Nakamura, F. Y. (2016). Initial Weekly HRV Response is Related to the Prospective Change in VO2max in Female Soccer Players. Int J Sports Med. doi:10.1055/s-0035-1569342
Nakamura, F. Y., Flatt, A. A., Pereira, L. A., Ramirez-Campillo, R., Loturco, I., & Esco, M. R. (2015). Ultra-Short-Term Heart Rate Variability is Sensitive to Training Effects in Team Sports Players. JOURNAL OF SPORTS SCIENCE & MEDICINE, 14(3), 602.
Results
At the current moment, results do not seem to offer a lot of insight regarding canoe training. They should be better presented.
We thank the reviewer, this is a study with canoeists, reporting reference values of canoeing training camps and the potential of applying non-invasive monitoring tools, that can help to avoid undesirable adaptations.
RESTQ-52 analysis should be presented.
We thank the reviewer. In the lines 179 to 183 we report data RESTQ-52
Training workload is not a result.
We thank the reviewer this comment. However, because one of the main objectives of the study were to analyse the load during the training camp and to analyse it. As training load was acceded by the perceived RPE and the duration of the training session, representing the internal and external workload (sRPE), is in our opinion important to show the real training content as it is express in table 1.
Discussion:
There is an abyss between results and interpretation. Data should be better explained and described, and then better interpreted. There is a lack of context also in discussion.
We thank the reviewer this comment. We rewrite this section aimed we hope that is more understandable now.
The aims of the study do not seem to be answered properly.
We thank the reviewer this comment. The text has been revised, which we hope will have satisfied the reviewer's doubts.

Round 2
Reviewer 3 Report
The authors do not show a willingness to improve the manuscript.